# Independent control of mean and noise by convolution of gene expression distributions

Karl P. Gerhardt[1], Satyajit D. Rao[1], Evan J. Olson[1], Oleg A. Igoshin [1,2,3,4] & Jeffrey J. Tabor [1,2✉]

Gene expression noise can reduce cellular fitness or facilitate processes such as alternative metabolism, antibiotic resistance, and differentiation. Unfortunately, efforts to study the impacts of noise have been hampered by a scaling relationship between noise and expression level from individual promoters. Here, we use theory to demonstrate that mean and noise can be controlled independently by expressing two copies of a gene from separate inducible promoters in the same cell. We engineer low and high noise inducible promoters to validate this result in *Escherichia coli*, and develop a model that predicts the experimental distributions. Finally, we use our method to reveal that the response of a promoter to a repressor is less sensitive with higher repressor noise and explain this result using a law from probability theory. Our approach can be applied to investigate the effects of noise on diverse biological pathways or program cellular heterogeneity for synthetic biology applications.

[1] Department of Bioengineering, Rice University, 6100 Main Street, Houston, TX 77005, USA. [2] Department of Biosciences, Rice University, 6100 Main Street, Houston, TX 77005, USA. [3] Center for Theoretical Biophysics, Rice University, 6100 Main Street, Houston, TX 77005, USA. [4] Department of Chemistry, Rice University, 6100 Main Street, Houston, TX 77005, USA. ✉email: jeff.tabor@rice.edu

Protein copy numbers vary within populations of genetically identical cells due to stochasticity in the molecular and cellular level events that impact gene expression[1–3]. This gene expression noise can be harmful, causing metabolic or physiological challenges, or reduce the reliability with which a cell can carry out a task[4–7]. Indeed, evolution appears to have optimized genomic locus[8], promoter architecture and sequence[9], transcription and translation rate[10], and selected for negative feedback[11,12] to decrease noise in the expression of toxic, complex forming, highly connected, and essential proteins[8,10,13–16]. On the other hand, evolution has also exploited noise as a means to regulate stress response[17–19], alternative metabolism[20], cell-fate determination[21], and pathways enabling cell populations to divide labor or hedge bets against unpredictable environments[22,23].

Gene expression noise can be considered to contain an intrinsic component, relating to stochasticity in the chemical reactions of gene expression processes, and an extrinsic component, relating to noise in global conditions or upstream factors[1,24,25]. At low numbers of expressed proteins, intrinsic noise dominates and inversely correlates with the mean. At higher protein copy numbers, extrinsic noise becomes dominant and uncorrelated with the mean.

Tools that specifically modulate noise in the expression of genes of interest are needed to study the impact of noise on biological processes. However, controlling noise independently of mean is a major challenge due to the coupling between these two parameters[9,13,14,26–28].

Several strategies for decoupling mean and noise have been reported. For example, researchers have altered promoter activation kinetics[9,29,30], operator site location and multiplicity[31,32], and introduced transcriptional feedback[33–35]. However, multiple strains must be engineered to achieve different noise levels for the same mean using these methods. Independent control of mean and noise in a single strain requires manipulation of two separate processes impacting protein copy number[36]. This result has been demonstrated by combining two small-molecule responsive regulators in a cascade[37–40], altering both the frequency and bias of promoter state switching[41], tuning both transcription and mRNA degradation rates[42], or using a time-varying input to independently control promoter activation frequency and transcription rate[43].

One important limitation to all of these previous approaches is that they rely upon genetic parts, circuits, or pathways that are native to or have been optimized to function in a particular organism. As a result, substantial re-engineering may be required to achieve the same results in each new organism of interest. Additionally, there may be fundamental limitations on the levels of gene expression and noise that can be achieved using these approaches. For example, a two-step cascade primarily allows control of extrinsic noise as it relies on transmission of noise from the upstream regulator to the output[39,40]. Likewise, modulation of promoter kinetics is expected to primarily modulate intrinsic noise[41] and is unlikely to be effective at high copy numbers.

Here, we show that combining the protein expression distributions from multiple promoters in a single cell is a generalizable and straightforward strategy to achieve robust and independent control of mean and noise over a wide area. To this end, we first use a simple theoretical model to reveal that the mean and noise of a population distribution can be independently controlled using two co-expressed and orthogonally regulated inducible promoters (IPs). We then implement this approach experimentally by constructing low and high noise generating IPs activated by the addition of two separate inducer molecules in *E. coli*. Next, we show that mean and noise of total gene expression can be manipulated using inducer combinations to control the activity of each IP. We characterize the steady-state behavior of

cells harboring the IP pair and present a simple mathematical model to predict mean and noise from inducer concentrations. Next, we show that our experimental gene expression profiles can be predicted with high accuracy by simulating convolutions between the distributions contributed by each IP. Finally, we use our approach to independently tune mean and noise in the expression of a bacterial transcriptional repressor and analyze how each affects the activity of a target promoter independently.

## Results

**Model for independent control of mean and noise from two IPs.** We considered two copies of a gene, encoding products $G_1$ and $G_2$, in a single cell. Here, the total amount of gene product, $G$, is $G_1 + G_2$. The mean ($\mu$) value of $G$ across a population of such cells is obtained from

$$\mu(G) = \mu(G_1) + \mu(G_2) \tag{1}$$

and the variance ($\sigma^2$) from

$$\sigma(G)^2 = \sigma(G_1)^2 + \sigma(G_2)^2 + 2\mathrm{Cov}(G_1, G_2) \tag{2}$$

where Cov is the covariance. If $G_1$ and $G_2$ are regulated such that their expression is stochastically independent and the covariation negligible, the noise ($\eta$, defined as the standard deviation ($\sigma$) divided by the mean) of $G$ is described by a weighted average of the noise from each source

$$\eta(G) = \left( \frac{\eta(G_1)^2 \mu(G_1)^2 + \eta(G_2)^2 \mu(G_2)^2}{(\mu(G_1) + \mu(G_2))^2} \right)^{\frac{1}{2}} \tag{3}$$

and the distribution of $G$ in the cell population is described by a convolution

$$p_G(g) = \int_{-\infty}^{\infty} p_2(g - g_1) p_1(g_1) dg_1 \tag{4}$$

where $p_1$, $p_2$, and $p_G$ are the probability density functions of $G_1$, $G_2$, and $G$, respectively. Conditions for stochastically independent $G_1$ and $G_2$ expression can be met if their dominant sources of noise are intrinsic or pathway specific[25].

Based on these results, we reasoned that mean and noise of $G$ could be independently controlled by regulating $G_1$ and $G_2$ expression from low and high noise IPs (Fig. 1a). In this approach, $\eta(G)$ can be varied between $\eta(G_1)$ and $\eta(G_2)$ while maintaining constant $\mu(G)$ by tuning the relative expression of $G_1$ and $G_2$ with ratios of IP inputs (Fig. 1b). $\eta(G)$ can be tuned this way at different values of $\mu(G)$ by controlling the absolute expression of $G_1$ and $G_2$ with the amount of IP inputs. We also reasoned that the distribution of $G$ can be predicted from a convolution of the distributions of $G_1$ and $G_2$ (Fig. 1c). One attractive feature of this approach is the direct relationship between tunability and the difference in noise produced by $G_1$ and $G_2$. Therefore, IPs that produce large differences in noise over the same range of means are desirable when implementing this method.

**Engineering a high noise promoter induced by AHL.** To engineer a system capable of tuning noise over a wide range, we designed two IPs that produce similar mean expression levels with very different noise values. First, we engineered a high noise IP that incorporates positive autoregulation through the 3-oxo-C$_6$-acylhomoserine lactone (AHL)-dependent transcriptional activator LuxR and its target promoter P$_{lux}$. Specifically, we expressed a bicistronic mRNA encoding the reporter gene superfolder green fluorescent protein (*sfgfp*) and *luxR* as the output of P$_{lux}$. To achieve high noise levels and strong inducibility, we generated a small library of variants of this IP with *luxR* ribosome binding sites (RBSs) of different strengths (Fig. 2a and

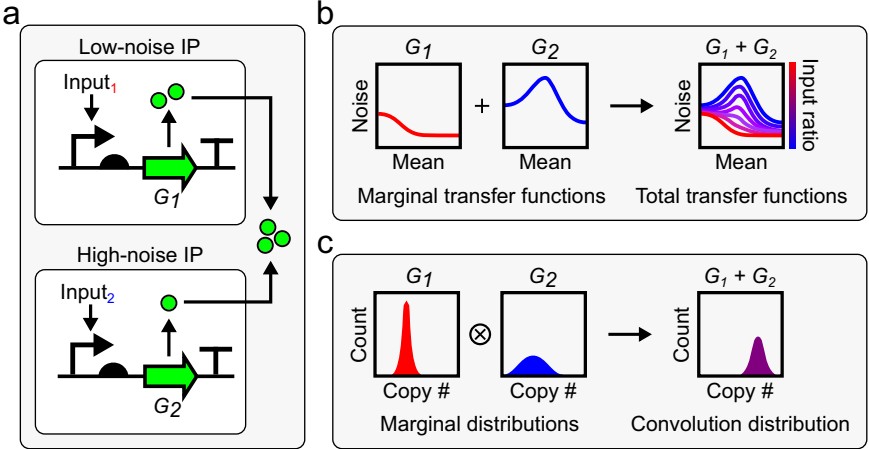

**Fig. 1 Summation of gene expression from low and high noise IPs. a** Two copies ($G_1$ and $G_2$) of the same gene are expressed from two independently regulated IPs: one that produces low noise distributions and one that produces high noise distributions. $G_1$ and $G_2$ sum inside cells. **b** Mean and noise of $G_1$ and $G_2$, separately, are functions of their respective IP's inputs (marginal transfer functions) and have single mean-noise trajectories. Mean and noise of the sum of $G_1$ and $G_2$ is a function of the amount and ratio of both IP's inputs (total transfer functions) and can be tuned within the area defined by their marginal transfer functions. **c** When $G_1$ and $G_2$ are summed, their marginal distributions form a convolution ($\otimes$).

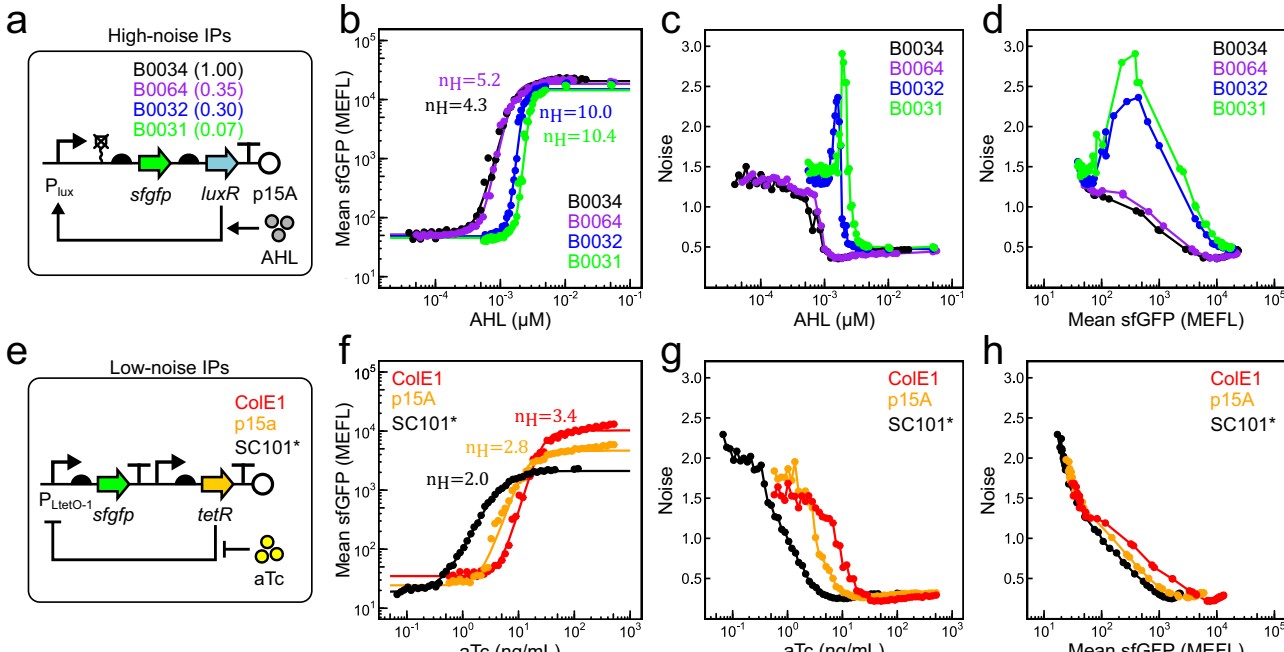

**Fig. 2 Engineering low and high noise IPs controlled by AHL and aTc. a** Library of high noise IPs based on LuxR-AHL-mediated positive transcriptional feedback. *LuxR* is encoded under RBSs of variable strength (indicated above the RBS) in different library members. Steady-state AHL-mean (**b**), AHL-noise (**c**), and mean-noise (**d**) transfer functions for high noise IP variants from **a**. **e** Library of low noise IPs based on TetR-aTc-inducible gene expression without feedback. Different plasmid origins on which this IP is introduced are indicated. Steady-state aTc-mean (**f**), aTc-noise (**g**), and mean-noise (**h**) transfer functions for origin of replication variants from **e**. Points within each group are single replicates collected from two separate experiments performed on 2 separate days. Smooth lines in **b** and **f** are fits to Hill functions with Hill coefficients ($n_H$) indicated.

Supplementary Fig. 1). Initially, we expected that stronger *luxR* RBSs would increase positive feedback strength by increasing the translational burst size of *luxR*. We believed this effect would result in more *luxR* expression per AHL molecule, a steeper AHL-mean transfer function, and higher noise at intermediate IP outputs[44–46]. To examine the performance of these high noise IP designs, we separately transformed each construct into bacteria, treated the resulting strains with different AHL concentrations, and measured the resulting sfGFP fluorescence distributions via flow cytometry. To our surprise, weaker *luxR* RBSs result in

increased steepness in the AHL-mean transfer function and higher noise in the AHL-noise and mean-noise transfer functions, respectively (Fig. 2b–d, Supplementary Fig. 1, and Supplementary Table 1).

To understand these effects, we developed a deterministic kinetic model of the high noise IP family (Supplementary Methods and Supplementary Table 2). This model details the binding interaction between LuxR and AHL, as well as positive transcriptional autoregulation by LuxR-AHL. While such a deterministic model cannot predict gene expression noise, it

allows us to analytically solve the steady-state response of the high noise IP family to AHL and identify design principles responsible for steepness of the transfer function. In particular, we find that this family of high noise IP designs is only sensitive to feedback when LuxR-AHL concentration is sensitive to LuxR fluctuations, i.e., when LuxR is limiting relative to AHL. At low *luxR* translation rates and intermediate AHL concentrations, the sfGFP output is bistable and exhibits hysteresis. Cell populations undergo abrupt jumps from low to high states in this regime, resulting in steep AHL-mean transfer functions (Supplementary Fig. 2).

Within this bistable window, we expected noise-driven transitions between states to result in high gene expression variability. To verify this prediction, we constructed a stochastic kinetic model of the positive feedback circuit and computed AHL-mean and AHL-noise transfer functions using Gillespie SSA simulations (Supplementary Methods and Supplementary Tables 3 and 4). Mean and noise values quantitatively match experimental values for all four RBSs (Supplementary Fig. 3). For weak RBSs, noise reaches a maximum at intermediate AHL concentrations, while stronger RBSs show monotonic decreases in noise, further demonstrating ultrasensitive circuit transitions due to bistability.

Taken together, these kinetic modeling results capture the performance of our high noise IP family and recapitulate the unexpected inverse relationship between *luxR* RBS strength and the magnitude of feedback in the circuit. Among the high noise IP variants we tested, the variant containing the B0031 RBS generated the highest overall noise while maintaining unimodality. With this variant (hereafter named $IP_h$), we have constructed a high noise IP that satisfies our design criteria.

**Engineering a low noise promoter induced by aTc**. To produce low noise gene expression distributions over a wide range of mean values, we designed an IP wherein *sfgfp* is expressed under control of the Tetracycline Repressor (TetR)-regulated $P_{Ltet-O1}$ promoter, with *tetR* expressed from a constitutive promoter on the same plasmid (Fig. 2e). We cloned the DNA encoding this IP into different plasmid backbones with ColE1 (50–70 copies/cell), p15a (20–30 copies/cell), and SC101* (3–4 copies/cell)[47] origins of replication (Fig. 2e). Initially, we hoped to find differences in output noise between origin of replication variants by virtue of the scaling between copy number and intrinsic noise, or by differences in plasmid copy number variability[25,47,48]. Mean output range (the difference between high and low states), detection threshold (inducer concentration at half output range), and steepness all increased with plasmid copy number (Fig. 2f and Supplementary Table 5). A deterministic kinetic model of the low noise IPs (Supplementary Table 6 and Supplementary Methods) capture these experimental behaviors (Supplementary Fig. 4) and provides an explanation for why their transfer functions become steeper with increasing plasmid copy number.

For all origin of replication variants, we also observed that the sfGFP noise decreases monotonically as a function of both inducer and sfGFP mean until reaching a noise floor of about 0.25 (Fig. 2g, h). At low induction, higher copy number variants produce lower noise but also correspondingly higher sfGFP mean, such that all variants collapse onto a similar initial trajectory. These behaviors suggest noise from this IP is dominated by intrinsic (at low to intermediate expression) and global extrinsic (at high expression) sources rather than transmitted noise from TetR or differences in copy number stringency between origins of replication. At intermediate induction, mean-noise transfer functions diverge slightly, with lower copy number variants decreasing more rapidly than higher copy number variants

(Fig. 2h). A stochastic kinetic model of this low noise IP (Supplementary Methods) recapitulates these experimental results, predicting monotonic decreases in noise as a function of aTc for all origin of replication variants, and lower noise at intermediate induction for lower copy number variants (Supplementary Fig. 3 and Supplementary Tables 7–9).

These kinetic models recapitulate the behavior of our low noise IPs and reveal that the observed performance differences that arise on different plasmid backbones are attributable to differences in repressor and promoter copy number. The SC101* variant generates the lowest overall noise and similar sfGFP mean output levels as $IP_h$. Thus, we renamed this variant $IP_l$ and carried it forward for further studies.

**Independent control of mean and noise with low and high noise IPs**. To demonstrate independent control of mean and noise by summing gene expression from low and high noise IPs, we co-transformed bacteria with plasmids encoding $IP_h$ and $IP_l$ (Fig. 3a). We exposed populations of the co-transformed bacteria to a $25 \times 25$ (625 total) panel of AHL and aTc concentrations and measured the sfGFP distributions by flow cytometry (Fig. 3b–d). We found that AHL-mean transfer functions are sigmoidal and shift higher with the level of aTc (Fig. 3b). This behavior is consistent with summation of sfGFP from $IP_h$ and $IP_l$. Both the AHL-noise and mean-noise transfer functions decrease non-monotonically, peak at intermediate AHL concentrations, and shift lower with the level of aTc (Fig. 3c, d). These properties are consistent with noise being determined by the relative contribution of $IP_l$ and $IP_h$ to total sfGFP (Eq. (3)). As intended, exposure to different inducer combinations produces an area in mean-noise space over which our system can be tuned (Fig. 3d). Thus, we can independently control mean and noise by summing gene expression from low and high noise IPs.

As predicted by our model, a wide range of sfGFP noise values can be achieved at virtually the same mean (Fig. 3d–g). At low total bacterial fluorescence levels, differences between distributions with similar mean and disparate noise are masked by *E. coli* autofluorescence (Fig. 3f). However, as sfGFP levels increase and the contribution of autofluorescence to total cellular fluorescence becomes negligible, differences between distributions with similar mean and different noise levels become dramatic (Fig. 3e, f). Thus, while we can tune mean and noise at low and high expression levels, detecting tunability at low mean requires analysis after autofluorescence subtraction.

We next determined whether Eqs. (1) and (3) could quantitatively recapitulate the behavior of this system. To that end, we adopted two phenomenological equations to describe mean and noise of each IP as a function of inducer, and fit their parameters to the experimental mean-noise data in Fig. 3 (Supplementary Methods). Following this approach, we observe close agreement between model predictions and experimental data, enabling accurate prediction of mean and noise from inducer concentrations and further supporting the hypothesis of additive gene expression from our two IPs. (Fig. 3b–d, Supplementary Fig. 5, and Supplementary Table 10).

**$IP_h$/$IP_l$ outperforms previous mean-noise control systems**. To our knowledge, no metric has been proposed to describe the ability of a genetically encoded system to independently tune gene expression mean and noise. The dynamic range, or ratio of output gene expression levels in the fully active versus fully inactive states, is a one-dimensional metric frequently used to quantify IP performance. However, mean and noise are tunable over a two-dimensional area[37–43].

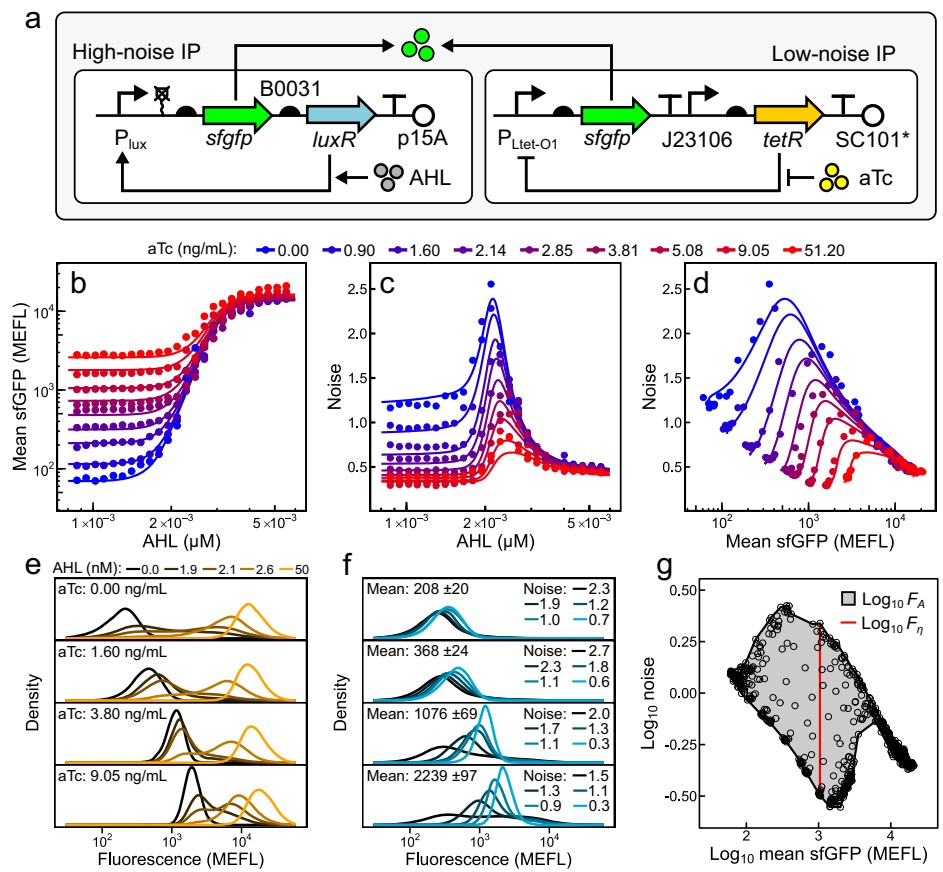

**Fig. 3 Independent control of mean and noise by summation of gene expression from $IP_h/IP_l$. a** Plasmids encoding $IP_h$ and $IP_l$ are co-transformed into *E. coli* and induced with AHL and aTc, respectively. sfGFP from each IP sum inside cells. Steady-state AHL-mean (**b**), AHL-noise (**c**), and mean-noise (**d**) transfer functions of cells harboring plasmids in **a** with exposure to combinations of AHL and aTc. Mean and noise values (points) and model fits (lines) shown are a subset (9 out of 25 aTc concentrations) selected for evenly spaced visualization. Selected fluorescence density estimates of cell populations induced with AHL and aTc (**e**) over a wide range of inducer concentrations, or **f** to a wide range of noise at virtually the same mean. **g** Concave hull (black line) of the complete mean-noise pointset (circles) used to calculate dynamic area and dynamic noise. Points and density estimates are single replicates collected from three separate experiments performed on 3 separate days.

To quantify the performance of mean-noise control systems, we developed metrics that we term dynamic area ($F_A$) and dynamic noise ($F_\eta$) (Methods). $F_A$ measures the fold-change in mean-noise area over which a system can be tuned, while $F_\eta$ measures the largest fold-change in noise a system can achieve at a constant mean. Practically, $F_A$ describes the capacity of a system to produce different combinations of both mean and noise, while $F_\eta$ captures the ability to modulate noise at a constant mean. We computed $F_A$ and $F_\eta$ using the experimental mean-noise dataset measured for our system and found values of 11.39 and 6.88, respectively (Fig. 3g). By this same analysis, we find that our system performs better than any previously described mean-noise control system (which range from 2.22 to 10.17 in $F_A$ and 1.61 to 4.66 in $F_\eta$) we are aware of, in any organism (Supplementary Fig. 6 and Supplementary Table 11).

Noise values among native *E. coli* genes range from 0.26 to 6.09 in a manner strongly dependent on the mean[13]. By comparison, our system can achieve noise values ranging from 0.318 to 2.18 at just a single mean (1045 MEFL, where $F_\eta$ is defined), making our system capable of tuning noise through a range which is physiologically relevant to *E. coli*.

**Convolution model predicts $IP_h/IP_l$ distributions.** We hypothesized that the distributions produced by the combined $IP_h/IP_l$

system could be predicted by simulating a convolution between distributions generated by $IP_l$ and $IP_h$ individually (Fig. 4a). To examine this hypothesis, we simulated each of the 625 experimental populations resulting from exposure to the AHL and aTc panel in Fig. 3 by summing randomly sampled fluorescence events between populations induced with only AHL and populations induced with only aTc (termed marginal distributions; Fig. 4b, Methods section). These simulated distributions show remarkable similarity to their experimental counterparts and frequently capture subtle, higher-order behaviors observed in experimental distributions such as skew and bimodality (Fig. 4b). While the simulated distributions are highly accurate overall, fluorescence levels are systematically overestimated in distributions with very low mean. This overestimation occurs because autofluorescence and basal sfGFP fluorescence (sfGFP fluorescence in the absence of inducer) are measured twice during the summation of two cell fluorescence events (Fig. 4a). Our distribution predictions are also less accurate when the AHL-induced population is near the inflection point of the AHL-mean transfer function (Fig. 3b).

We quantified similarity between each pair of experimental and predicted distributions using the Bhattacharyya coefficient[49] ($c_B$), a metric ranging from 0 to 1 measuring overlap between two probability distributions (Fig. 4b, c, Methods section). The average $c_B$ for all populations is remarkably high, at 0.92 with a standard deviation of 0.098. However, due to the previously described effect of

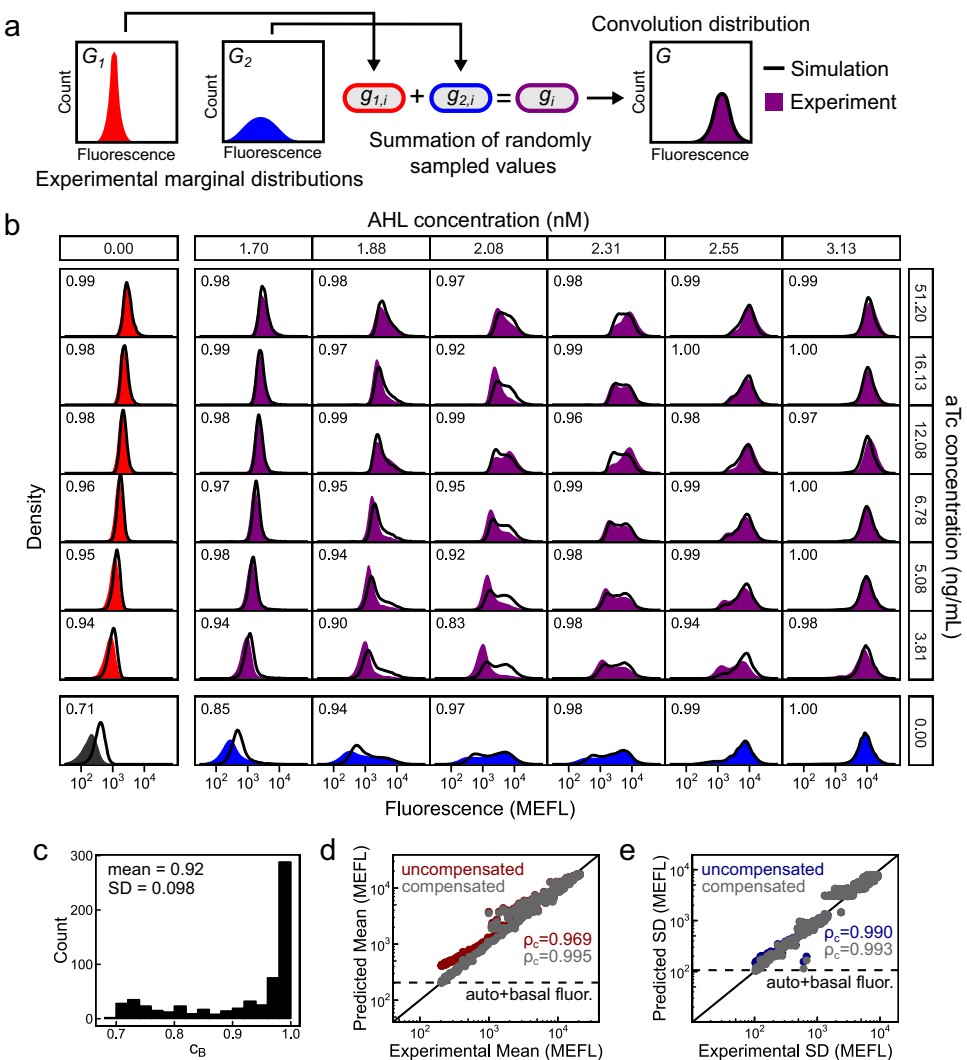

**Fig. 4 Convolution predicts gene expression distributions from IP$_h$/IP$_l$. a** Workflow for simulating convolution distributions. Fluorescence events from two experimental marginal distributions are randomly sampled and summed together. Summed events form a simulated convolution distribution which can be compared with an experimental counterpart. **b** Experimental (filled histograms) and simulated (black lines) fluorescence distributions of cell populations induced with combinations of AHL (vertically aligned, 7 of 25 shown) and aTc (horizontally aligned, 7 of 25 shown). Simulations were performed using populations with no AHL induction (first column) and no aTc induction (bottom row) as marginal distributions. Bhattacharyya coefficients ($c_B$) for each experimental-simulated distribution pair is listed in the upper left of each subpanel. **c** Distribution of $c_B$ for all 625 pairs of experimental and simulated distributions. **d, e** Comparison of mean and standard deviation (sd) of experimental and simulated distributions. Equivalence line (black line) and Lin's concordance coefficients $\rho_c$ are shown. Uncompensated predicted values measure autofluorescence and basal sfGFP fluorescence (dashed line) twice due to the summation in **a**. Compensation is described in the Methods section. Experimental density estimates are single replicates collected from three separate experiments performed on 3 separate days.

overestimating autofluorescence and basal sfGFP expression, $c_B$ shows strong correlation with expression mean below ~1000 MEFL (Supplementary Fig. 7). Mean and standard deviation of simulated distributions show strong concordance with their experimental counterparts ($\rho_c$ of 0.969 and 0.990 respectively) and this concordance is further improved ($\rho_c$ of 0.995 and 0.993 respectively) after compensation for overestimated autofluorescence and basal expression (Fig. 4d, e, Methods section). Overall, this approach of simulating convolutions between two experimental marginal distributions enables simple and accurate prediction of the total fluorescence distributions generated by our system.

**Repressor noise decreases steepness of promoter response.** The transcription factor-promoter transfer function is the quantitative relationship between transcription factor expression level and target promoter activity. It has previously been shown that the shape, including the steepness, of a transcription factor-promoter transfer function can strongly depend on the levels and context of transcription factor expression[50–54]. Based on previous experiments measuring the impact of noise on biological processes[29,37,43], we hypothesized that increasing noise in transcription factor expression would produce less steep transcription factor-promoter transfer functions. We used IP$_h$/IP$_l$ to characterize the effect of repressor noise on mean expression from a target promoter. To that end, we first fused PhlF[AM 50], a TetR family repressor from *Pseudomonas fluorescens*, to the C-terminus of sfGFP on both IP$_h$ and IP$_l$. We then co-transformed *E. coli* with plasmids expressing the modified IP$_h$ and IP$_l$ along with an output plasmid carrying *mCherry* expressed

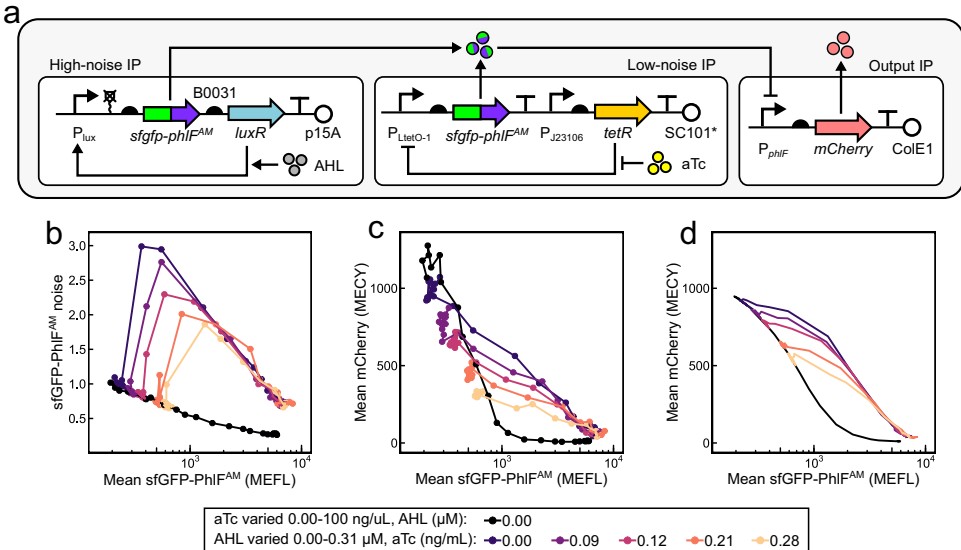

**Fig. 5 Noise modulates repressor activity on a target promoter. a** Schematic representation of plasmids used to control noise of a transcriptional regulator and monitor its output. PhlF$^{AM}$ is fused to the C-terminus of sfGFP and expressed from IP$_l$ and IP$_h$. PhlF$^{AM}$ activity is monitored via mCherry expression from the P$_{phlF}$ promoter on the output IP. **b** Steady-state sfGFP-PhlF$^{AM}$ mean-noise transfer functions of cells harboring plasmids in **a** with exposure to combinations of AHL and aTc. **c** Mean mCherry expression as a function of mean sfGFP-PhlF$^{AM}$. **d** LOTUS model prediction of mean mCherry as a function of sfGFP-PhlF$^{AM}$. Points represent single replicates collected on a single day.

under the PhlF$^{AM}$ repressed P$_{PhlF}$ promoter (Fig. 5a). We grew these bacteria under different combinations of AHL and aTc, allowed sfGFP and mCherry levels to reach steady-state, and quantified both fluorescent proteins by flow cytometry (Fig. 5b, c and Supplementary Figs. 8–10). As before, mean-noise transfer functions decrease non-monotonically with respect to AHL induction, peak at intermediate AHL concentrations, and shift lower upon addition of aTc (Fig. 5b). Conversely, induction with only aTc results in comparatively low noise, which monotonically decreases with higher mean (Fig. 5b). As a result, the system achieves different sfGFP-PhlF$^{AM}$ noise while maintaining the same mean by applying different amounts and ratios of AHL and aTc. Mean mCherry fluorescence decreases in response to mean sfGFP-PhlF$^{AM}$ in a manner that strongly depends on noise (Fig. 5c and Supplementary Fig. 10). When induced with AHL, mean mCherry begins to decrease at lower mean sfGFP-PhlF$^{AM}$, but requires dramatically higher mean sfGFP-PhlF$^{AM}$ to become fully repressed compared to when induced with aTc (Fig. 5c and Supplementary Fig. 10). Moreover, induction with combinations of AHL and aTc generates intermediate levels of this effect. Noise, therefore, makes the PhlF-promoter transfer function overall less steep, increases apparent PhlF$^{AM}$ activity at low mean, and decreases apparent PhlF$^{AM}$ activity at high mean. This striking result demonstrates how a single transcription factor can generate vastly different population level behaviors depending on the details of its expression.

We applied a simple law from probability theory (Methods section) to predict mean mCherry values from probability distributions of sfGFP-PhlF$^{AM}$ (Supplementary Methods, Supplementary Fig. 11, and Supplementary Table 12). While predicted mean mCherry values show a qualitative agreement with the data (Fig. 5d), strong cell autofluorescence signal in the fluorescence distributions at low expression levels likely undermines a more quantitative agreement.

## Discussion

In nature, gene duplication has been proposed as a mechanism to resolve a tradeoff between expression noise and environmental

responsiveness[55]. This hypothesis was recently validated for a pair of duplicated transcription factors in *Saccharomyces cerevisiae*, where one member of the pair exhibits low noise and is constitutively expressed, while the other exhibits high noise and is induced by environmental stress[56]. This strategy appears to have evolved to minimize transcription factor noise under normal growth conditions while also allowing activation of a stress response pathway under stress-inducing conditions. Based on the results of this study, we hypothesize that gene duplication may also allow cells to adjust gene expression noise in order to increase fitness in environments where low noise is beneficial and in other environments where high noise is beneficial.

Unlike previous approaches to modulating mean and noise in gene expression, our method does not require the use of a particular genetic part or circuit, a specific mechanism of noise reduction or amplification, or a particular host organism. Rather, it only requires the availability of orthogonal low and high noise generating IPs in an organism of interest and the ability to co-transform them into a single cell.

Though several mean-noise control systems have been reported, there had been no method available to benchmark their performance against one another. Here, we propose dynamic area and dynamic noise to quantify the performance of mean-noise control systems. These metrics capture the magnitude of the mean-noise area and the noise accessible at a constant mean, respectively. Like dynamic range, these metrics are independent of the absolute values of mean and noise. By comparing these two variables, one could distinguish systems that are tunable in both dimensions from systems that are primarily tunable in just a single dimension.

Our model based on summation of gene expression from two IPs accurately predicts population-level mean and noise of fluorescence distributions from inducer concentrations. The success of this approach supports the hypothesis for additive gene expression from our two engineered sources and may be adapted to describe future implementations of this method.

We can predict total gene expression distributions by simulating convolutions between experimental IP$_l$ and IP$_h$ distributions. While the predictions are systematically overestimated

when autofluorescence and basal reporter gene fluorescence dominate (at low mean), they are otherwise surprisingly accurate and able to capture detailed population features that would not be predicted following a parametric approach. Going forward, our predictions would benefit from a method to reduce or deconvolve autofluorescence from flow cytometry measurements. While convolution predictions require advance measurement of marginal distributions, the number of convolutions that can be predicted combinatorially increases with number of marginal distributions measured. We imagine this predictive ability could be utilized to forward engineer desired convolution distributions, including their higher moments and noise types, from existing promoter libraries[57–63], constitutive or otherwise. Likewise, our method could be used to combinatorially increase the number of gene expression distributions achievable with a constitutive promoter library by expressing the same gene from combinations of two or more library members.

Our results indicate that regulated promoters respond to their cognate transcription factors at lower mean expression levels but require higher mean expression levels to saturate as transcription factor noise increases. As a result, a noisy transcription factor has higher apparent activity than a less noisy transcription factor at low expression levels, while the opposite is true at high levels. These findings could be used to anticipate the effect of changing the distribution of a transcriptional regulator or create design principles for predicting and programming the shape of transcriptional dose-response curves.

Our method could be used to study other noise-dependent biological phenomena such as transient stochastic resistance[19,64], persister cell formation[17,18], stochastic differentiation[21], noise-induced cooperative behaviors[22,23], or gene circuit stability[65] in cell populations. Such studies could provide a greater experimental basis for the fitness advantages conferred by noise or be used to create design principles for engineering desirable cell population behaviors using noise-driven processes.

Gene expression convolution could also be used to engineer noisy phenotypes into populations of living cells. Example behaviors include genetically identical populations that automatically differentiate into specified ratios of cell sub-types[66], form Turing-type patterns using activator and inhibitor morphogens with similar diffusion rates[67], or stochastically lyse in order to release enzymes to enable the population to metabolize complex agricultural feedstocks[68]. Taken together, gene expression convolution is a simple strategy for studying and controlling gene expression noise in a wide range of organisms and biological pathways.

## Methods

**Molecular biology**. Plasmids used in this study are listed in Supplementary Table 13. Plasmid maps are shown in Supplementary Fig. 12. DNA assembly was performed by the Golden Gate method[69], and cloning was performed in strain NEB 10β (New England Biolabs). A PhlF mutant (PhlF$^{AM}$) with improved repression activity was amplified from the genome of strain sAJM.1506[50] and used in this study.

## Cell growth and chemical induction

*IP characterization and convolution experiments*. Experiments were performed in strain MG1655 in M9 media + 100 mM HEPES (pH 6.6) at 37 °C and 250 RPM of shaking. Media was supplemented with ampicillin (50 μg/mL), spectinomycin (100 μg/mL), and chloramphenicol (35 μg/mL) as appropriate to maintain plasmids.

Frozen preculture aliquots of each experimental strain were generated by growing transformants to exponential phase (OD$_{600}$ ≈ 0.1), adding glycerol to 18% (v/v), recording OD$_{600}$, and freezing 100 μL aliquots in PCR tubes at −80 °C.

Panels of chemical inducer concentrations were prepared by diluting varying amounts of AHL (Sigma-Aldrich, K3007) in media, and aTc (Takara, 63130) in 100% ethanol, in wells of 96-well plates. Inducer concentrations in each panel well were prepared to 200X the desired final concentration. Panel plates were sealed

with adhesive foil, stored at −30 °C, and warmed to room temperature before experiments.

Experimental cultures were prepared by diluting a volume of preculture in media to achieve a cell density of OD$_{600}$ = 2 × 10$^{-5}$. Culture media was distributed among wells of 24-well plates (1 mL/well) and supplemented with the desired chemical inducer concentration by multichannel pipetting solution from chemical inducer panels. Culture plates were then sealed with adhesive foil and grown for 6 h to OD$_{600}$ ≤ 0.3, after which time they were iced for ≥15 min and measured by flow cytometry.

*Transcription factor noise experiments*. Experiments were performed as described above but in LB media. LB media was found to be necessary due to high metabolic burden from mCherry expression.

**Flow cytometry**. Flow Cytometry was performed with a BD FACScan flow cytometer outfitted with blue (488 nm, 30 mW) and yellow (561 nm, 50 mW) solid-state lasers (Cytek) and FlowJo CE (7.5.110.7) acquisition software. sfGFP fluorescence was measured in the FL1 channel with a 510/20 nm emission filter, and mCherry fluorescence was measured in the FL3 channel with a 650 nm long-pass filter. Event rates of 1000–3500 events/s were used, and all events were captured until 20,000 events occurred within an SSC-FSC area characteristic of the strain. Calibration beads (Spherotech, RCP30-5A) were measured at the end of each cytometry session. Flow cytometry files were processed using FlowCal[70]. A gate fraction of 0.3 was used to gate events in the SSC and FSC channels, and FL1/FL3 arbitrary fluorescence units were calibrated to MEFL/MECY using calibration bead data collected during each respective cytometry session (Supplementary Fig. 13).

**Population mean and variance calculation**. Flow cytometry data for each sample was analyzed using a custom Python script that calculates arithmetic mean and noise from fluorescence distributions. The script first trims a small number of outlier observations which can heavily influence sample noise. Trimming is performed by first calculating a smooth estimation of the probability density function corresponding to the log-fluorescence distribution of the sample via kernel density estimation. The range of fluorescence values to keep is then determined by identifying the points nearest to the median at which the density estimate falls below a 0.5% threshold. Sample histograms that display the trimming effect are generated and assessed to ensure the trimming functions as expected. Mean and variance of a likewise analyzed, untransformed MG1655 sample are then subtracted from the mean and variance of experimental samples.

**System performance analysis**. The concave hull of log$_{10}$ transformed mean-noise pointsets was used to calculate both $F_A$ and $F_\eta$. R packages alphahull (2.2) and sp (1.4-1) were used to find concave hulls and convert hulls to polygon objects, respectively. Alpha parameter was chosen manually. We report log$_{10}(F_A)$ as the area of the polygon defined by the hull, and log$_{10}(F_\eta)$ as the length of the longest vertical chord spanning the hull.

**Convolution distribution simulation and analysis**. Fluorescence events were randomly sampled ($n = 50,000$) from populations that received no AHL and summed with likewise sampled events from populations that received no aTc. To simulate the population which received neither AHL nor aTc, events were sampled from that population twice and then summed. The three highest and lowest fluorescence events were eliminated before summation to remove extreme outliers.

Experimental and simulated fluorescence distributions were converted to density estimates, $p_e$ and $p_s$, respectively, to calculate Bhattacharyya coefficients:

$$c_B = \int_0^\infty \sqrt{p_e(x)p_s(x)}dx \tag{5}$$

where $x$ is FL1 fluorescence (MEFL).

Compensation of predicted mean and standard deviation was performed by subtracting the mean and variance of autofluorescence and basal sfGFP fluorescence from the uncompensated predicted mean and variance. Agreement between log$_{10}$-scale experimental and simulated mean and standard deviation was measured by Lin's concordance correlation coefficient ($\rho_c$)[71].

**Model for transcriptional output of a noisy regulator**. Mean transcriptional output from the P$_{PhlF}$ promoter was modeled using the Law of The Unconscious Statistician (LOTUS):

$$\mu(c) = \int_0^\infty f(x)p_e(x)dx \tag{6}$$

where mean mCherry expression, $\mu(c)$, of a population is found through a single-cell relationship, $f(x)$, between FL1 and FL3 fluorescence (Supplementary Methods) and the experimental FL1 probability density estimate $p_e(x)$. Model predictions in FL1 and FL3 were converted to sfGFP and mCherry by autofluorescence subtraction.

**Statistical analysis**. Points and density estimates throughout the text are single replicates collected over one to three separate experiments as indicated in figure legends. Standard errors on model fits are shown in the Supplementary Information.

**Reporting summary**. Further information on research design is available in the Nature Research Reporting Summary linked to this article.

## Data availability
Plasmid sequences generated in this study have been deposited in the GenBank database under the accession numbers listed in Supplementary Table 13. Processed flow cytometry data generated during this study has been deposited on GitHub (https://github.com/taborlab/NoiseControl)[72]. Any other relevant data can be obtained from the authors upon reasonable request.

## Code availability
Custom code used in this study has been deposited on GitHub (https://github.com/taborlab/NoiseControl)[72].

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

## Acknowledgements

We thank Dr. Joel Moake for the use of his flow cytometer. This material is based upon work supported by the National Science Foundation under Grant No. CAREER 1553317 (PI: J.J.T.), the Office of Naval Research under Grant No. N00014-18-1-2611 (PI: J.J.T.), the Welch Foundation under Grant No. C-1856 (PI: J.J.T.), the National Science Foundation under grant No. MCB-1616755 (PI: O.A.I.), and the Welch Foundation under grant No. C-1995 (PI: O.A.I.).

## Author contributions

K.P.G. and J.J.T. conceived the project. J.J.T. and O.A.I. supervised the project. K.P.G. designed and constructed plasmids, performed experiments, and analyzed data. E.J.O. developed and wrote the code for calculation of sample mean and noise. S.D.R. developed and analyzed kinetic models of IPs. K.P.G. and E.J.O. developed the phenomenological model for IP mean and noise. K.P.G. developed and conducted performance analysis and convolution simulation technique. K.P.G., S.D.R., O.A.I., and J.J.T. wrote the manuscript.

## Competing interests

The authors declare no competing interests.
