## [Peer Review File · Nature Communications]

Reviewers' Comments:

Reviewer #1:

Remarks to the Author:

This manuscript by Gerhardt, Rao, Olson, Igoshin, and Tabor describes a genetic circuit scheme to control mean and noise of gene expression semi-independently. The authors present a design using two inducible promoters with different inherent noise levels that, when used together, can achieve a large range of expression mean and noise combinations through variation in two inducer concentrations. The authors also present a quantitative model of the system that recapitulates the experimental data very closely. The manuscript is well-prepared and clearly written. Although other publications do report independent control of noise and mean, this study is by far the most extensive and is the system with the greatest control that I am aware of, making it an exciting advance for the field.

Major Concerns:

1. The authors openly acknowledge that noise control at low expression does not work well. The authors attribute this to autofluorescence and/or leaky expression, which sounds reasonable. However, it would be helpful to include data showing autofluorescence on the same scale and an attempt to correct for it. For example, Fig. 4d/e compensates for this effect. It would be illustrative to perform the same compensation for other major figures (e.g. Fig. 3g).

2. I had some trouble understanding Fig. 5 and the discussion surrounding it. The caption and line 308 of the text both mention different combinations of AHL and aTc, but only an aTc gradient is shown in the legend above Fig. 5 b/c. If AHL isn't added to any of the curves, what is the role of the high noise IP in this case? Also, I may be misunderstanding something, but if I look at a sfGFP-PhIF MEFL level of 2×10^3 the mCherry expression mean is different for all induction combinations, but the expression noise is the same. I'm fairly confused as to how to interpret this but think readers may generally benefit from a clearer explanation of these results.

Minor Concerns:

3. Schematics in Fig. 2 and Fig. S3 have the order of sfGFP and luxR flipped. Were the constructs different for these experiments?

4. I'm wondering about the physiological relevance of the noise values in Fig. 3. Can the authors comment on how these compare to well-known noisy promoters to give context to these noise values.

5. In Fig 5d, are all the mCherry expression distributions unimodal or are some bimodal. If the authors could include this information either with a comment or supplemental figure it will help to interpret these results. Fig. S10 shows this, but it's hard to tell the distribution because the data points are on top of each other. It might be useful to either see the data points colored for density or to show histograms for a few selected examples.

6. Fig. 5 caption typo "single replicates"

7. In line 396, the authors say that high noise in a transcription factor results in promoters responding to lower concentrations, but with less sensitivity. This wording is a bit confusing to me. Can you expand on how you are defining sensitivity here?

Reviewer #2:

Remarks to the Author:

Summary: The authors develop a synthetic approach to independently control the mean and noise of an expressed protein in *E. coli*. This is an important problem because mean and noise in biochemical systems are normally coupled to each other. Correspondingly, perturbations to the noise will simultaneously affect the mean (or vice versa), which makes it difficult to test the impact of noise on biological systems. While previous studies have dealt with the same problem, the method proposed in the present study is relatively simple and general. In particular, it uses two copies of the same gene with different inducible promoters. These promoters have different mean-noise characteristics and when used in superposition, can achieve tunable noise levels for a given mean protein concentration. The authors also provide a simple theoretical approach to predict experimental protein distributions for the given system. Finally, the authors use their approach to

study the effect of varying noise levels on a downstream gene, which is repressed by the inducible gene. This reveals that noise in the repressor decreases the sensitivity of the downstream promoter. Overall, the experimental approach presented in this manuscript is relevant and may be important in the future to better interrogate the effects of noise on biological systems. I have a several comments below, especially relating to the theoretical part of the work. I believe these comments should be addressed before the manuscript can be published.

-The authors neglect the covariation term between the two gene product concentrations G1 and G2, arguing that if G1 and G2 are "regulated", the covariation should be negligible. By default I would suspect that those genes still show non-negligible variation due to the presence of extrinsic variability. I believe that neglecting these correlations may be justifiable for the sake of simplicity but this should be stated clearly. Also, the authors argue that the covariation can be accounted for but I don't see how this can be done without performing additional measurements of the covariation (e.g., using a dual reporter setup).

-The description/justification of the theoretical approach is poor. There are many places where some of the assumptions come out of the blue. Moreover, it is currently a patchwork of different stochastic / deterministic models but these individual parts often seem ad hoc and unrelated to each other. It is also not clear what some of these efforts add to the study. I would recommend the authors to revisit the theoretical part of this work and streamline it. Here are a few concrete examples:

* it seems that extrinsic noise is incorporated in the LuxR model but not in the TetR model? What is the standard deviation that is used in the lognormal distribution accounting for extrinsic variability? How were the parameters that are affected by extrinsic variability chosen?

* I don't see how the stochastic TetR model relates to the deterministic one. How were the rates / reaction volumes chosen to have them consistent with each other?

* Where does equation 27 come from and why is a detailed model for the convolution approach even needed? The individual distributions (and correspondingly mean and variance) can be measured experimentally and then convolved to obtain the intermediate solutions, right?

* Similarly around 30, I could not follow the discussion about how intrinsic noise of LuxR is captured.

* Then in 32, a phenomenological function is introduced to replace some of the noise terms with some other terms. I think given all the assumptions that go into obtaining these equations, the fits (e.g., as shown in Fig3 b,c) could also be replaced just by a standard polynomial fit (or something alike). Again, I don't understand why this more detailed model is needed at all and what it adds to the study.

- Why are Figure 3b,c,d only shown for AHL induction?

Minor comments:

-For my taste, the LOTUS is a bit over-emphasized. In this logic, every calculation of a statistical moment (e.g., mean / variance) would have to be justified via LOTUS, since these are also expectations over functions of random variables. Explaining more clearly what has been done would probably be more helpful than invoking LOTUS.

-line 756: "ofrom" should be "from"

REVIEWER COMMENTS

Reviewer #1 (Remarks to the Author):

This manuscript by Gerhardt, Rao, Olson, Igoshin, and Tabor describes a genetic circuit scheme to control mean and noise of gene expression semi-independently. The authors present a design using two inducible promoters with different inherent noise levels that, when used together, can achieve a large range of expression mean and noise combinations through variation in two inducer concentrations. The authors also present a quantitative model of the system that recapitulates the experimental data very closely. The manuscript is well-prepared and clearly written. Although other publications do report independent control of noise and mean, this study is by far the most extensive and is the system with the greatest control that I am aware of, making it an exciting advance for the field.

Major Concerns:

1. The authors openly acknowledge that noise control at low expression does not work well. The authors attribute this to autofluorescence and/or leaky expression, which sounds reasonable. However, it would be helpful to include data showing autofluorescence on the same scale and an attempt to correct for it. For example, Fig. 4d/e compensates for this effect. It would be illustrative to perform the same compensation for other major figures (e.g. Fig. 3g).

Author response:

We thank the reviewer for flagging the lack of clarity here. While it is difficult to visually see noise tunability at low mean from the distributions alone (e.g. Fig. 3f), tunability is more apparent (e.g. Fig 3d) after computing mean and noise and subtracting autofluorescence (Methods: 'Population mean and variance calculation'). We perform this autofluorescence subtraction throughout the manuscript for summary statistics of sfGFP and mCherry.

Figure 4 requires a different type of compensation because autofluorescence and basal sfGFP expression are effectively measured twice during the summation of two cell fluorescence events (Fig 4a). To compensate for this specific effect, we subtract the autofluorescence and basal sfGFP expression component from the mean and standard deviation of the simulated distributions. We have clarified the source of this effect and the way we are compensating for it in the figure (Fig 4 d,e), figure legend, and Methods (Convolution distribution simulation and analysis).

2. I had some trouble understanding Fig. 5 and the discussion surrounding it. The caption and line 308 of the text both mention different combinations of AHL and aTc, but only an aTc gradient is shown in the legend above Fig. 5 b/c. If AHL isn't added to any of the curves, what is the role of the high noise IP in this case? Also, I may be misunderstanding something, but if I look at a sfGFP-PhIF MEFL level of 2×10^3 the mCherry expression mean is different for all induction combinations, but the expression

noise is the same. I'm fairly confused as to how to interpret this but think readers may generally benefit from a clearer explanation of these results.

Author response:

We thank the reviewer for pointing out the need for more clarity in our presentation of Figure 5. We did vary both aTc and AHL in Figure 5 b-d and have updated the figure to be clearer about how this was done. Specifically, the legend now conveys which inducer was varied and which was held constant for each color group.

The reviewer points out that at constant 2×10^3 sfGFP-PhIF mean, mCherry mean varies with inducer combination, while expression noise is unchanged. We are not sure if the reviewer is referring to sfGFP-PhIF (original panel b) or mCherry (original panel c) expression noise, but we agree that mCherry noise curves mostly overlap between inducer combinations. We believe the confusion here might be that mCherry noise is plotted with respect to mCherry mean (not sfGFP-PhIF mean). In other words, for the same mCherry mean, mCherry noise is relatively unaffected by inducer combination and hence noise in sfGFP-PhIF. This panel and its discussion (that sfGFP-PhIF noise propagation to mCherry appears to be dwarfed by other sources of noise) has been moved to the supplementary information to avoid this confusion and because noise propagation is beyond the focus and scope of this figure.

Minor Concerns:

3. Schematics in Fig. 2 and Fig. S3 have the order of sfGFP and luxR flipped. Were the constructs different for these experiments?

Author response:

We thank the reviewer for pointing this out. We have corrected the order of the sfGFP and luxR genes in the schematic.

4. I'm wondering about the physiological relevance of the noise values in Fig. 3. Can the authors comment on how these compare to well-known noisy promoters to give context to these noise values.

Author response:

The most extensive characterization of gene expression noise in native *E. coli* genes we are aware of comes from Taniguchi et al. *Science*, 2010 (10.1126/science.1188308). In this paper, the highest reported CV is 6.086 (at 0.06 mean protein copies/cell) while the lowest is 0.265 (at 89.684 mean protein copies/cell). At the mean where our system's noise is most tunable (1045 MEFL, where F_η is defined) the maximum and minimum achievable CV is 2.18 and 0.318. We have added this discussion to the Results section of the main text.

5. In Fig 5d, are all the mCherry expression distributions unimodal or are some bimodal. If the authors could include this information either with a comment or supplemental figure it will help to interpret these results. Fig. S10 shows this, but it's hard to tell the distribution because the data points are on top of each other. It might be useful to either see the data points colored for density or to show histograms for a few selected examples.

Author response:

We thank the reviewer for this suggestion. We have added Supplemental Figure 9., showing sfGFP-PhIF and corresponding mCherry fluorescence distributions for several aTc and AHL induction concentrations. mCherry expression distributions display bimodality (with some symmetry to corresponding sfGFP distributions) at intermediate AHL concentrations, but are otherwise unimodal.

6. Fig. 5 caption typo "single replicates"

Author response:

We thank the reviewer for catching this typo. It has been corrected.

7. In line 396, the authors say that high noise in a transcription factor results in promoters responding to lower concentrations, but with less sensitivity. This wording is a bit confusing to me. Can you expand on how you are defining sensitivity here?

Author response:

We agree with the reviewer the use of 'sensitivity' here is not well defined. The aim was to refer to the overall slope/steepness of the transcription factor-promoter transfer function. We have replaced its use with more descriptive language in the text.

Reviewer #2 (Remarks to the Author):

Summary: The authors develop a synthetic approach to independently control the mean and noise of an expressed protein in E. coli. This is an important problem because mean and noise in biochemical systems are normally coupled to each other. Correspondingly, perturbations to the noise will simultaneously affect the mean (or vice versa), which makes it difficult to test the impact of noise on biological systems. While previous studies have dealt with the same problem, the method proposed in the present study is relatively simple and general. In particulate, it uses two copies of the same gene with different inducible promoters. These promoters have different mean-noise characteristics and when used in superposition, can achieve tunable noise levels for a given mean protein concentration. The authors also provide a simple theoretical approach to predict experimental protein distributions for the given system. Finally, the authors use their approach to study the effect of varying

noise levels on a downstream gene, which is repressed by the inducible gene. This reveals that noise in the repressor decreases the sensitivity of the downstream promoter. Overall, the experimental approach presented in this manuscript is relevant and may be important in the future to better interrogate the effects of noise on biological systems. I have a several comments below, especially relating to the theoretical part of the work. I believe these comments should be addressed before the manuscript can be published.

-The authors neglect the covariation term between the two gene product concentrations G1 and G2, arguing that if G1 and G2 are "regulated", the covariation should be negligible. By default I would suspect that those genes still show non-negligible variation due to the presence of extrinsic variability. I believe that neglecting these correlations may be justifiable for the sake of simplicity but this should be stated clearly. Also, the authors argue that the covariation can be accounted for but I don't see how this can be done without performing additional measurements of the covariation (e.g., using a dual reporter setup).

Author response:

We agree with the reviewer that being regulated does not itself justify independence or neglecting covariation. Rather, we meant to assert that independence could, in principal, be achieved by engineering the right type of regulation. Our justification for this assertion: 'IP expression may be independent if their dominant sources of noise are intrinsic and/or pathway specific (Raser & O'Shea, Science 2005)' originally came late in the paragraph. We have moved this justification to immediately follow the assertion. We have also modified the language to reflect a more hypothetical tone.

We agree additional measurements, such as with a dual reporter setup, would be required to directly quantify covariation. While one could perform these measurements if covariation could not be ignored, we do not do so in this study and have therefore removed the text corresponding to this point.

-The description/justification of the theoretical approach is poor. There are many places where some of the assumptions come out of the blue. Moreover, it is currently a patchwork of different stochastic / deterministic models but these individual parts often seem ad hoc and unrelated to each other. It is also not clear what some of these efforts add to the study. I would recommend the authors to revisit the theoretical part of this work and streamline it. Here are a few concrete examples:

Author response:

We thank the reviewer for the suggestion to streamline the theoretical part of this work.

The deterministic and stochastic molecular (now 'kinetic') models were constructed to explain the differences in behavior between the individual IPs developed in this study.

Namely, they explain why we see different levels of noise for the same mean with different plasmid copy numbers (TetR IPs) and translation rates (LuxR IPs).

We have incorporated additional background and description of the kinetic models in the Supplementary Methods. We also introduce a section to the supplementary methods relating parameters between the deterministic and stochastic kinetic models for each IP type. Based on this analysis, we re-fit the deterministic model for LuxR IPs using initial values and constraints from related parameters in the stochastic version of the model. This update to the deterministic model produces a qualitatively similar fit with parameters values in close agreement with those of the stochastic model.

* it seems that extrinsic noise is incorporated in the LuxR model but not in the TetR model? What is the standard deviation that is used in the lognormal distribution accounting for extrinsic variability? How were the parameters that are affected by extrinsic variability chosen?

Author response:

The simplest TetR-aTc stochastic model was able to replicate noise characteristics as well as mean dose-response for increasing promoter copy numbers. Therefore, we did not consider extrinsic noise for this model. The only drawback we find of not modeling extrinsic noise is that the noise floor (CV for fully activated IP) of our model is lower than the data. We considered adding extrinsic noise in the form of variability in parameters such as promoter copy numbers, however, given that noise from the model matched the data qualitatively, we chose not to explore this further.

On the other hand, simulations of our stochastic AHL-LuxR IPs did not match experimental noise characteristics without extrinsic noise. Thus, we added extrinsic noise to our simulations. We chose the smallest set of parameters that could vary from cell to cell: protein dilution rate (depends on cell doubling time/cell size), mRNA decay rate, basal transcription rate, and plasmid copy number. A CV of 0.15 was used for lognormal distributions to account for extrinsic variability. Other sets of parameters could also be chosen in principle. However, since our goal for the stochastic model was to match and explain observed mean and noise as a function of dose (AHL) and RBS strength (translation rate), rather than make predictions, we did not explore the set of parameters that would contribute to extrinsic noise any further.

* I don't see how the stochastic TetR model relates to the deterministic one. How were the rates / reaction volumes chosen to have them consistent with each other?

Author response:

We thank the reviewer for highlighting this lack of clarity. The stochastic models are more detailed than the deterministic models. Thus, how the stochastic models relate to the deterministic models is not obvious. Therefore, we have added an analysis

comparing deterministic steady state expressions for the stochastic model to the analytical solutions of the simple deterministic models (Eqs. (13)-(15), and Eqs. (19)-(23)). We update our model descriptions with an explicit section comparing parameters between deterministic and stochastic kinetic models of both LuxR and TetR IPs.

* Where does equation 27 come from and why is a detailed model for the convolution approach even needed? The individual distributions (and correspondingly mean and variance) can be measured experimentally and then convolved to obtain the intermediate solutions, right?

Author response:

Equation 28 (formerly 27) is the Hill-function, commonly used to fit gene expression dose-response curves (termed mean-inducer transfer functions in this study).

As the reviewer suggests, a discrete approach using two experimentally measured distributions is possible. Indeed, we show this approach in Fig. 4 d,e. Our 'detailed' (now 'phenomenological') approach enables continuous prediction of mean and noise as a function of inducer concentrations, which we feel is a valuable demonstration.

However, we agree with the reviewer that the phenomenological model was more complicated than need be and overemphasized. We have greatly simplified the approach and its discussion by adopting a single equation (Eq. 29) to describe the noise of both IPs as a function of inducer concentration, with terms for constant, 1/mean, and peak behaviors. We favor this to a polynomial approach, as those terms individually capture behaviors associated with extrinsic, intrinsic, and transmitted noise. We also note that the functional form for this approach has literature precedent (Pedraza and van Oudenaarden, *Science*, 2005 (10.1126/science.1109090)).

* Similarly around 30, I could not follow the discussion about how intrinsic noise of LuxR is captured.

Author response:

We have greatly simplified the phenomenological approach and its discussion by adopting a single equation (Eq. 29) to describe the noise of both IPs as a function of inducer concentration. Intrinsic noise in this case is captured by a 1/mean term scaled by a constant.

* Then in 32, a phenomenological function is introduced to replace some of the noise terms with some other terms. I think given all the assumptions that go into obtaining these equations, the fits (e.g., as shown in Fig3 b,c) could also be replaced just by a standard polynomial fit (or something alike). Again, I don't understand why this more detailed model is needed at all and what it adds to the study.

Author response:

The phenomenological approach enables continuous prediction of mean and noise as a function of inducer concentrations, which we feel is a valuable demonstration. We have greatly simplified the approach (see above). We favor this to using a polynomial, as the noise terms individually capture behaviors associated with extrinsic, intrinsic, and transmitted noise. We also note that the functional form for this approach has literature precedent (Pedraza and van Oudenaarden, *Science*, 2005 (10.1126/science.1109090)).

- Why are Figure 3b,c,d only shown for AHL induction?

Author response:

Figure 3b,c,d are plotted with respect to AHL and colored by ATC concentration. An AHL vs aTc heatmap visualization of the dataset and model predictions is also included in supplementary Fig. 5.

Minor comments:

-For my taste, the LOTUS is a bit over-emphasized. In this logic, every calculation of a statistical moment (e.g., mean / variance) would have to be justified via LOTUS, since these are also expectations over functions of random variables. Explaining more clearly what has been done would probably be more helpful than invoking LOTUS.

Author response:

We thank the reviewer for their perspective on this section. We have significantly trimmed the discussion of LOTUS modelling in the main text, relying more on the methods and SI to explain the details if readers are curious.

-line 756: "ofrom" should be "from"

Author response:

We thank the reviewer for catching this typo. It has been corrected.

Reviewers' Comments:

Reviewer #1:

Remarks to the Author:

The authors have addressed my concerns.

Reviewer #2:

Remarks to the Author:

The authors have largely addressed my comments. The description of the theoretical parts and how they relate to each other is more clear now. I have a few more (relatively minor) comments:

-As mentioned, the description of the models is more clear now. One thing that is still unclear is why the deterministic model is necessary in the first place, since a full stochastic model is developed and used later. A bit more motivation in the main text why the authors need/want to take this intermediate step would be helpful.

-I could not find a complete description of how the parameters of the stochastic model were chosen. It is discussed how these parameters can be related to the four parameters of the deterministic model, but this does not fully specify the (larger number of) parameters in the stochastic model. Presumably, the specific choices of these parameters will in general affect the noise that is predicted from the stochastic model. It seems that section "Corresponding parameters between..." in the Supplement is supposed to address this point, but the description was not sufficient for me to fully understand what has been done.

-Similarly, it was not immediately clear to me how the parameter relationships from Table 4 are derived. I suggest to either include these derivations or explain more clearly how they can be obtained.

Typos:

-Line 182: inducer concentration "at" half output range.

-eq (20): LuxR_0 is later called [LuxR]_0

GENERAL COMMENTS FROM AUTHORS

We thank the editor for continued consideration of our manuscript and the reviewers for their comments. We have addressed the remaining reviewer comments, which we feel has added additional clarity and context for the reader. We have provided details of the changes we have made below. We believe the manuscript is now ready for publication.

REVIEWER COMMENTS

Reviewer #1 (Remarks to the Author):

The authors have addressed my concerns.

Author response: Thank you for providing us a thoughtful review, which helped improve our manuscript.

Reviewer #2 (Remarks to the Author):

The authors have largely addressed my comments. The description of the theoretical parts and how they relate to each other is more clear now. I have a few more (relatively minor) comments:

-As mentioned, the description of the models is more clear now. One thing that is still unclear is why the deterministic model is necessary in the first place, since a full stochastic model is developed and used later. A bit more motivation in the main text why the authors need/want to take this intermediate step would be helpful.

Author response: We thank the reviewer for this suggestion. The deterministic models allow us to find analytical solutions for the steady-state behaviors of our engineered IPs, and to gain insight into their design principles. For example, we use a deterministic model to reveal that for strong *luxR* ribosome binding sites (RBSs), our high-noise IP always exhibits one stable steady-state where the output increases smoothly as AHL increases. Alternatively, for weak RBSs, our high-noise IPs exhibit two stable steady states and one unstable steady state, making AHL responses extremely sharp, and therefore noisy (Supplementary Fig 2). These effects occur because our high noise IP is only sensitive to transcriptional feedback when LuxR is limiting relative to AHL. The stochastic model primarily serves to validate our conclusions from the deterministic model. We have added language to the main text to clarify the purpose of the deterministic models (lines 148-151).

-I could not find a complete description of how the parameters of the stochastic model were chosen. It is discussed how these parameters can be related to the four parameters of the deterministic model, but this does not fully specify the (larger number of) parameters in the stochastic model. Presumably, the specific choices of these parameters will in general affect the noise that is predicted from the stochastic model. It seems that section "Corresponding parameters between..." in the Supplement is supposed to address this point, but the description was not sufficient for me to fully understand what has been done.

Author response: Thank you for raising this concern. In response, we have added a description to the Supplementary Methods (Corresponding parameters between the stochastic and deterministic models) describing how stochastic model parameters are specified based on four deterministic model parameters.

-Similarly, it was not immediately clear to me how the parameter relationships from Table 4 are derived. I suggest to either include these derivations or explain more clearly how they can be obtained.

Author response: We have added intermediate steps for deriving the parameter relationships in Table 4 to the Supplementary Methods (Corresponding parameters between the stochastic and deterministic models).

Typos:

-Line 182: inducer concentration "at" half output range.

-eq (20): LuxR_0 is later called [LuxR]_0

Author response: We thank the reviewer for pointing out these typos, they have been corrected.

Reviewers' Comments:

Reviewer #2:

None